# The Relationship Between Autoimmune Disorders and Multiple Sclerosis: Clinical Insights and Therapeutic Approaches

**DOI:** 10.3390/brainsci15060588

**Published:** 2025-05-30

**Authors:** Magdalena Iwan, Weronika Wójtowicz, Jakub Milczarek, Natalia Wyroba, Zuzanna Wydrych, Olga Falger, Michalina Rzepka, Tomasz Chmiela, Mateusz Toś, Joanna Siuda

**Affiliations:** 1Students’ Scientific Association, Department of Neurology, Faculty of Medical Sciences in Katowice, Medical University of Silesia, 40-752 Katowice, Polandnika240398@gmail.com (W.W.); jakubmilczarek95@wp.pl (J.M.); nmcwyroba@gmail.com (N.W.); zuzia.wydrych2412@gmail.com (Z.W.); olga.falger345@gmail.com (O.F.); 2Department of Neurology, Faculty of Medical Sciences in Katowice, Medical University of Silesia, Medykow 14 Street, 40-752 Katowice, Poland; michalinaj93@gmail.com (M.R.); tchmiela@sum.edu.pl (T.C.); jsiuda@sum.edu.pl (J.S.)

**Keywords:** multiple sclerosis, autoimmune diseases, thyroid disease

## Abstract

Background: Multiple sclerosis (MS) and autoimmune diseases (AIDs) share immunological underpinnings, leading to frequent co-occurrence. This study investigated the prevalence of AIDs among Polish patients with MS (PwMSs) and its potential effects on disease characteristics. The aims were to compare clinical and demographic characteristics between PwMSs with and without coexisting AIDs. Methods: A retrospective analysis was conducted on data from 580 PwMSs who were treated at the Department of Neurology, University Clinical Center in Katowice, Poland, between February 2018 and August 2023. Variables analyzed included age, sex, MS type, disease duration, treatment, Expanded Disability Status Scale (EDSS) scores, thyroid-stimulating hormone (TSH), and vitamin D3 serum concentrations. Results: AID was identified in 16.9% of PwMSs (n = 98). Compared with PwMSs without AIDs, PwMSs with AIDs exhibited significantly higher mean age (44.61 ± 11.40 vs. 42.24 ± 12.27 years; *p* = 0.0151), longer disease duration (10.77 ± 6.72 vs. 9.56 ± 7.19 years; *p* = 0.0102), and higher EDSS scores (2.97 ± 1.43 vs. 2.89 ± 1.84; *p* = 0.0261). Among PwMSs, the prevalence of AIDs was significantly higher in females (20.24%) compared to males (8.13%; *p* = 0.0022), and strongly associated with the relapsing-remitting MS subtype (*p* = 0.0352). Autoimmune thyroid diseases were markedly the most prevalent in PwMSs (hypothyroidism 7.24%). Conclusions: PwMSs with AIDs exhibit distinct characteristics, including older age, increased disease duration, and greater disability. Thyroid disorders are notably the most prevalent AIDs among PwMSs. These findings underscore the intricate interplay between AIDs and MS and highlight the necessity for further research into their long-term impact.

## 1. Introduction

Autoimmune diseases (AIDs) represent a significant challenge in contemporary medicine, owing to their increasing prevalence and chronic clinical course. AIDs constitute a heterogenous group of disorders characterized by the presence of autoantibodies or autoreactive T cells directed against host tissues or organs. Suppression of the immune response can influence the course of disease. At present, more than 80 distinct autoimmune disorders have been identified. Although AIDs may target any organ system, the tropism for the endocrine, nervous systems, and connective tissue is especially manifested [1,2,3].

Autoimmune mechanisms have been demonstrated to play a pivotal role in the pathogenesis of demyelination. Multiple sclerosis (MS) is an immune-mediated neurodegenerative disease of the central nervous system, characterized by inflammatory demyelination with axonal transection [4,5,6]. There are three major forms of MS based on the course of the disease: relapsing-remitting MS (RRMS), secondary progressive MS (SPMS), and primary progressive MS (PPMS) [6,7,8]. With growing understanding of processes underlying the pathophysiology of MS, division into distinct clinical subtypes is insufficient, and MS as a disease continuum based on biological processes is more accurate [9]. This new classification framework focuses on the underlying mechanisms driving the disease, such as inflammation and neurodegeneration, as opposed to the manner in which symptoms present themselves [9]. In Poland, the incidence rate of MS in 2019 was 11.6 per 100,000 inhabitants, while the prevalence rate was 244.9 per 100,000 inhabitants [10]. Current trends indicate a shift in the prevalence of MS towards older age groups, with the highest prevalence estimated in the sixth decade of life [11]. The female population is affected 2–3 times more frequently than the male population, although male patients tend to exhibit more severe clinical signs and symptoms [12]. While the precise causes of MS remain unclear, both genetic and environmental factors are thought to contribute to the development of MS [13,14].

MS is widely regarded as an immune-mediated condition. Nevertheless, the question of whether inflammation is the primary driver or a secondary epiphenomenon remains controversial. Two mechanistic frameworks have emerged in recent years. The classical ‘outside-in’ paradigm attributes early perivascular T-cell infiltration and demyelination to aberrant adaptive immunity, whereas the ‘inside-out’ concept proposes that a primary oligodendrocyte or axonal insult triggers a secondary immune response [15,16,17]. Converging high-field magnetic resonance imaging (MRI) and neuropathological studies also demonstrate that demyelination of the cortical and deep grey matter can equal or even exceed white-matter involvement, thus challenging the historical view of MS as a purely white-matter disease [18,19]. It is imperative to reconcile these perspectives in order to interpret the comorbidity patterns, including autoimmune co-occurrence, observed in MS.

The incidence of MS is influenced by a range of factors, including geographical latitude [7]. Among the environmental contributors, Epstein–Barr virus infection and a deficiency in 25-hydroxyvitamin D (henceforth referred to as “vitamin D3”) are thought to play significant roles in triggering this process, but further research on this topic is needed, as the pathogenesis seems to be more complicated [20,21].

AIDs broadly share pathological mechanisms, where genetic predispositions must be coupled with environmental triggers to provoke an immune response [22]. It has been established that AIDs are more likely to develop in individuals with pre-existing autoimmune conditions, prompting an investigation into how the prevalence of coexisting AIDs among patients with MS (PwMSs) contrasts with the prevalence of those diseases in the general population [23]. Vitamin D3 deficiency has been identified as a risk factor for the development of both MS and AIDs, through its role in various anti-inflammatory and immunomodulatory processes. Vitamin D3 receptors (VDRs) are also found on cells and tissues of the immune system (e.g., thymus and spleen), which may explain the immunomodulatory effects of calcitriol [21]. Dendritic cells are of particular significance within this field. These antigen-presenting cells are responsible for generating an immune response and inducing immunological tolerance [21]. However, supplementation has not demonstrated a notable effect on enhancing clinical outcomes in PwMSs [24,25].

Despite the numerous investigations conducted, the association between MS and an increased prevalence of AIDs remains unresolved, and the findings remain inconclusive. Several studies have highlighted an increased prevalence of AIDs in PwMSs, thereby suggesting the potential for a shared pathogenic mechanism between MS and other autoimmune disorders [26,27]. However, contradictory evidence proposes that this correlation may be overstated, possibly stemming from the heightened frequency of symptom reporting by PwMSs, rather than a true increase in comorbidity [28,29]. The variability in these results underscores the need for further research to clarify the relationship.

This study aimed to determine the prevalence of coexisting AIDs among PwMSs and to analyze the relationship between clinical and demographic characteristics among PwMSs regarding the potential occurrence of AIDs.

## 2. Materials and Methods

### 2.1. Study Population

Our retrospective comparative study was conducted on the medical records of 580 adult PwMSs who were hospitalized in the Department of Neurology at the University Clinical Centre of the Medical University of Silesia in Katowice, Poland, between February 2018 and August 2023. The diagnosis of MS was established in accordance with the McDonald criteria from 2010 [30] or the modified McDonald criteria from 2017 [31]. The study population comprised only those PwMSs for whom complete clinical data were available. From an initial cohort of 840 patients, 260 (31.0%) were excluded due to missing key information, including confirmed MS diagnosis, Expanded Disability Status Scale (EDSS) score, MS type, disease duration, or the presence of comorbidities. The evaluation of subjects was conducted using the EDSS scores obtained at the time of the most recent hospitalization [32]. Ethical committee approval was not required, due to its retrospective comparative nature of this study and data anonymization.

### 2.2. Data Collection

The analysis encompassed a range of clinical and demographic parameters, including gender, age, age at onset, form of MS (RRMS, SPMS, PPMS, and Rapidly Evolving Severe (RES-RRMS)), disease duration, prescribed disease-modifying treatments (DMTs), and the presence of coexisting AIDs. The categorization of DMT was conducted in accordance with the classification system for drugs, which encompasses the division into two distinct categories: Low-Efficacy Treatment Agent (LETA) which included interferons, glatiramer, teriflunomide, fumarates, and High-Efficacy Treatment Agent (HETA) which included sphingosine-1-phosphate-receptor (S1PR) modulators, natalizumab, anti-CD20 monoclonal antibodies, alemtuzumab, and cladribine [33]. Comorbidity data were extracted from patients’ medical records.

Additionally, the study investigated the presence of vitamin D3 and thyroid-stimulating hormone (TSH) in the serum. According to the standards of the laboratory where the vitamin D3 levels were determined, results in the range of 30–50 ng/mL were considered normal, values < 30 ng/mL as below the norm, and levels above 100 ng/mL were regarded as toxic. We considered TSH serum levels between 0.4 and 4.0 mlU/L as the reference range. Patients with TSH serum levels < 0.4 mlU/L were classified as below the norm, while those with TSH serum levels > 4.0 mlU/L were categorized as above the standard value [34].

### 2.3. Statistical Analysis

All statistical calculations were performed using Statistica 13.0 (TIBCO Software Inc., Palo Alto, CA, USA). The presentation of the data included means and standard deviations for quantitative variables, and percentages for qualitative ones. The normality of the distribution of quantitative variables was assessed using the Shapiro–Wilk test. The Student’s *t*-test was used to compare quantitative variables with a normal distribution, while the Mann–Whitney U test was used for those with a non-normal distribution. The chi-square test was utilized to compare qualitative variables. To explore multivariate relationships within the dataset, we employed Principal Component Analysis (PCA), Partial Least Squares (PLS) regression, and multivariable regression models. PCA was used as an unsupervised method to explore the overall structure of the clinical data and potential clustering patterns related to the presence of AIDs. Standardized continuous variables included age, age at MS diagnosis, disease duration, EDSS, MS subtypes, sex, and treatment status. PLS regression was applied as a supervised learning method to predict the presence of AIDs (binary outcome: 1 = AIDs present, 0 = AIDs absent). This method allowed the inclusion of collinear and potentially correlated predictors. We used two latent components, and the model’s predictive performance was assessed using the area under the receiver operating characteristic curve (AUC). Feature importance was derived from the magnitude of PLS coefficients. Additionally, two multivariable regression models were constructed. A logistic regression was used to assess the association between clinical variables and the presence of AIDs. Separately, a linear regression model was applied to evaluate the effect of covariates on the EDSS. The level of statistical significance was set at *p* < 0.05.

## 3. Results

### 3.1. General Description of the Study Group

The study group comprised a total of 580 PwMSs. The cohort was predominantly female, with 431 women (71.71%) and 170 men (28.29%). The mean age of PwMSs was 42.88 ± 12.56 years (mean ± SD), ranging from 18 to 73 years. The average age at MS diagnosis was 33.63 ± 11.22 years, with a mean disease duration of 9.82 ± 7.58 years. RRMS was the most prevalent form (80.86%; n = 469), followed by SPMS (8.28%; n = 48), PPMS (7.58%; n = 44), and RES-RRMS (3.28%; n = 19). The median EDSS score was 2.50 (IQR: 1.25–3.75). The majority of PwMSs (51.90% n = 301) were on LETA. A total of 216 individuals (37.24%) were on HETA, while 10.86% of the PwMSs (n = 63) included in the study were not undergoing any form of treatment. Among the group of PwMSs studied, the most used DMTs were dimethyl fumarate (28.97%; n = 168), followed by teriflunomide (11.03%; n = 64), ocrelizumab (10.34%; n = 60), and ofatumumab (4.66%; n = 27) (Table 1).

The mean vitamin D3 serum level was 34.52 ± 27.20 ng/mL, with 47.59% of PwMSs (n = 237) having below-normal levels, 50.20% (n = 250) having normal levels, and 2.29% (n = 11) having toxic levels. The mean TSH level was 2.06 ± 4.54 µIU/mL. The majority of PwMSs exhibited normal TSH levels (94.75%; n = 469), while 2.42% (n = 12) showed low levels and 2.83% (n = 14) exhibited elevated levels (Table 2).

### 3.2. AIDs in the Study Group

A total of 98 PwMSs (16.90%) had at least one co-occurring AID. Among these, 93 PwMSs had one AID (16.03%), and 5 PwMSs had two AIDs (0.86%) (Figure 1).

The most prevalent autoimmune disorders observed were autoimmune thyroid diseases, present in 77 PwMSs (13.28%), including hypothyroidism (7.24%; n = 42) and Hashimoto’s disease (4.18%; n = 24), hyperthyroidism (1.72%; n = 10), and other/unspecified thyroid disorders (0.17%; n = 1). Other AIDs included Type 1 Diabetes (1.55%; n = 9), inflammatory bowel diseases (1.03%; n = 6), and psoriasis (0.52%; n = 3). The distribution of AIDs in PwMSs is presented in Table A1 in the Appendix A.

### 3.3. Comparison Between PwMSs with and Without Co-Occurring AIDs

The distribution of gender was found to demonstrate a significant association with the presence of AIDs, with a higher prevalence observed in females (20.24%; n = 85) compared to males (8.13%; n = 13; *p* = 0.0022) within the group of PwMSs. The mean age was higher in the group of PwMSs with co-occurring AIDs than in the group without them (44.61 ± 11.40 years vs. 42.24 ± 12.27 years; *p* = 0.0151). In a cohort of PwMSs with co-occurring AIDs, the mean age of MS diagnosis was 34.38 ± 10.08 years. Conversely, PwMSs who did not have any autoimmune comorbidities had a mean age of MS diagnosis 33.27 ± 11.29 years. The findings indicate no statistically significant difference between the two groups regarding the age of MS diagnosis. Patients with AIDs demonstrated significantly prolonged disease duration when compared to those without concomitant (10.77 ± 6.72 years vs. 9.56 ± 7.19 years; *p* = 0.0102).

Furthermore, the cohort of PwMSs with co-occurring AIDs exhibited a higher mean EDSS score in comparison to PwMSs without additional AIDs (3.00 IQR: 1.75–4.25 vs. 2.50, IQR 1.25–3.75; *p* = 0.0261). Kernel density plots were used to visualize the distribution of EDSS in PwMSs with and without AIDs (Figure 2).

Comparing PwMSs with and without AIDs, no statistically significant differences in DMTs were found (*p* = 0.4192). In the separate groups of RRMS, SPMS, PPMS, and RES-RRMS patients, the percentage of individuals with autoimmune comorbidities was, respectively, 18.12% (n = 85), 16.67% (n = 8), 11.36% (n = 5), and 0.00% (n = 0) (*p* = 0.0352) (Table 3).

### 3.4. The Results of Vitamin D3 and TSH Levels in PwMSs with and Without Co-Occurring AIDs

There were no significant differences in vitamin D3 levels between PwMSs with and without AIDs (*p* > 0.05). A comparison between PwMSs with and without AIDs revealed that the group with AIDs exhibited higher TSH levels (3.71 ± 10.68 vs. 1.72 ± 1.04, *p* = 0.041). PwMSs with AIDs demonstrated a significantly lower prevalence of normal TSH levels compared to those without AIDs, with only 84.52% of the AIDs group maintaining normal TSH levels compared to 96.84% of individuals without AIDs (*p* < 0.0001).

### 3.5. Comparison of Groups in Relation to the Prevalence of Thyroid Disease

Among women with MS, 17.62% (n = 74) were found to have coexisting thyroid disease. In contrast, the prevalence of concomitant thyroid disease among men with MS was significantly lower, at 1.88% (n = 3) (*p* < 0.0001). The mean age was found to be significantly higher in the cohort of PwMSs with thyroid disease, averaging 46.66 ± 12.20 years, compared to 42.32 ± 12.17 years in those without thyroid disease (*p* = 0.007). The mean age of MS diagnosis differed between groups; however, this difference did not reach statistical significance (*p* > 0.05). Specifically, the average age at diagnosis for PwMSs without thyroid disease was 33.32 ± 11.22 years, whereas it was 35.61 ± 11.06 years for those with thyroid disease. Additionally, the duration of MS was significantly longer among PwMSs with thyroid disease, with a mean duration of 11.62 ± 8.16 years, in contrast to 9.54 ± 7.46 years in those without thyroid involvement (*p* = 0.029).

Concomitant thyroid disease was observed most frequently among patients with SPMS, where it was present in 14.58% of cases (n = 7). Similarly, the occurrence of thyroid disease in patients with RRMS was noted at 14.29% (n = 67). Conversely, only 3 patients (6.82%) with PPMS were diagnosed with thyroid conditions. Moreover, no cases of thyroid disease were identified within the RRMS-RES subgroup (Table 3).

The mean serum vitamin D3 levels were also comparable between the groups, showing no significant disparity. PwMSs without thyroid disease exhibited mean levels of 34.83 ± 28.64 ng/mL, while those with thyroid disease had levels of 32.48 ± 14.84 ng/mL. Among PwMSs without thyroid disease, 205 (47.56%) exhibited vitamin D3 deficiency, 215 (49.88%) had normal levels, and 11 (2.55%) presented with toxic concentrations. In comparison, among those with thyroid disease, 32 (47.76%) were deficient in vitamin D3, 35 (52.24%) had normal levels, and none exhibited toxic concentrations.

### 3.6. Multivariate Analyses of AIDs Risk in PwMSs

PCA was applied to examine whether underlying clinical patterns could differentiate PwMSs with and without AIDs. The analysis aimed to explore underlying patterns in the dataset and identify clustering based on AIDs status. The first two principal components captured the majority of the variance. Visual inspection of the PCA plot (Figure 3) revealed partial but not complete separation between PwMSs with and without AIDs, suggesting that while multivariate structure exists, substantial clinical overlap remains across groups.

PLS regression was performed to predict AIDs status using the same clinical variables. This supervised method accounted for collinearity among predictors and extracted latent components that best explained variance in the response. The model achieved an AUC of 0.74, indicating moderate discriminative ability. Variable importance derived from PLS coefficients showed that the most influential predictors were progressive MS subtypes (SPMS and PPMS), longer disease duration, and older age. In contrast, gender and treatment status contributed minimally to the classification model.

In multivariable logistic regression, progressive MS subtypes were significantly associated with the presence of AIDs. Specifically, SPMS and PPMS increased the odds of AIDs occurrence (SPMS: OR = 4.18, *p* < 0.001; PPMS: OR = 3.47, *p* < 0.001). Age was also positively associated with AIDs status (OR = 1.04, *p* = 0.011). No significant associations were found for gender or treatment status. The linear regression model for EDSS showed that disease duration (β = 0.09, *p* < 0.001), MS subtypes (SPMS: β = 1.28, *p* < 0.001; PPMS: β = 1.42, *p* < 0.001), and age (β = 0.04, *p* < 0.001) were significant predictors of disability severity. Sex and treatment status did not significantly contribute to EDSS outcomes.

## 4. Discussion

The coexistence of MS and AIDs has been a subject of discussion for a considerable number of years [26,27,29]. The present research is a retrospective comparative study performed on the largest Polish cohort of people with MS (PwMSs) regarding co-existence of AIDs. To the best of our knowledge, no studies involving larger patient populations have been conducted up to the present date. Moreover, an attempt was made to ascertain the prevalence of AIDs within the PwMSs population in comparison to that observed in the general population.

### 4.1. Impact of Coexisting AIDs on MS Course

In our study, a significant portion of PwMSs was affected by AIDs (98 out of 580 individuals; 16.90%). The study revealed that PwMSs with AIDs had slightly higher levels of disability measured with EDSS than those PwMSs without AIDs.

The findings of this study indicate a slightly higher EDSS and longer disease duration in PwMSs who also have co-existing AIDs. These findings directly address the long-standing debate over whether inflammation (“outside-in”) drives neurodegeneration in MS or represents a secondary response to a primary oligodendrocyte insult (“inside-out”). The modest clinical difference, in conjunction with an unaltered relapse rate, suggests that systemic autoimmunity may be an epiphenomenon rather than a disease-accelerating factor [15,16,17]. In view of the recent evidence for extensive grey-matter demyelination [18,19], it is clear that further research is required to develop integrative models that accommodate both mechanisms along the MS continuum.

However, a study conducted in France involving sixty-six patients with both MS and inflammatory bowel diseases (IBD) found that MS had a milder course (mean EDSS score 2.5) in these patients compared to those with MS alone (mean EDSS score 3.0) [35]. In a study by Sahraianet et al. conducted at 1700 PwMSs, only twenty-four of them had concurrent AIDs. Furthermore, these patients also exhibited a lower average EDSS score compared to the control group (1.62 ± 1.12 vs. 3.33 ± 1.89). The authors hypothesized that the coexistence of AIDs might have a positive impact on the course of MS, potentially leading to increased tolerance of antibodies [36]. It may suggest that the concurrent autoimmune activity might prevent the full focus of the immune system on neurological tissues, leading to a less aggressive disease course. In comparison with the studies referenced above, the increased EDSS observed in the present research may be explained by several factors, including variations in the timing of patient diagnoses, their advanced average age, or the impact of diverse DMTs. These findings differ from another Polish study, which examined a smaller cohort of 27 patients with AIDs out of 381 PwMSs and reported no significant difference in EDSS scores [37]. Notably, discrepancies in EDSS outcomes for MS patients with comorbid AIDs may stem from measurements taken at different time points among heterogeneous populations with varied disease durations. Notwithstanding the methodological constraints, EDSS remains a valuable reference point for the comparison of disease progression in MS patients with and without AIDs.

MS, AIDs, and their co-occurrence have been observed to be more prevalent in women than in men [4,12,22,23]. In our study cohort, women comprised the majority of PwMSs (20.24%), both among those with MS alone and those with concurrent AIDs, aligning with existing literature. In comparison, only 8.13% of men were affected [38]. The literature indicates that MS predominantly affects women [4], with a consistent rise in its incidence among females and no corresponding decrease in males [39,40]. Women are also more susceptible to other AIDs, such as thyroiditis, rheumatoid arthritis (RA), and lupus erythematosus systemic US (SLE). Immune responses, including those against antibodies, are typically more robust in women than in men [41,42].

In our study, PwMSs with concurrent AIDs were statistically older than those without AIDs. In the French study, the authors described PwMSs and IBD. The median age at diagnosis of IBD was found to be 31 years (range 21–40 years) and without MS also being 31 years (range 23–38 years), which, contrary to our study, demonstrates no association between autoimmune diseases and older age in PwMSs [35]. Those differences might reflect variations in study cohorts or methodologies. There is a lack of other studies that clearly indicate differences in the age of PwMSs with and without coexisting AIDs.

When considering the form of MS in the clinical description, a higher prevalence of RRMS is noted among the group of PwMSs with AIDs. In the group of PwMSs and AIDs, 86% were diagnosed with RRMS, whereas RRMS was identified in 79% of patients without AIDs. This distribution may suggest a milder course of MS in individuals with AIDs, although further research is needed to confirm this hypothesis. The low prevalence of these MS types could either reflect the overall predominance of RRMS or a genuine influence of AIDs on disease progression [43].

### 4.2. Effect of Immunomodulatory Therapies on AID Course and Morbidity

The development of AIDs in patients with MS may occur subsequent to a neurological diagnosis, with the administration of immunomodulatory therapies playing a contributory role. In our study, we analyzed the types of DMTs taken by PwMSs, categorizing them into HETA and LETA groups. Statistical analysis revealed no significant differences in the prevalence of AIDs among PwMSs using diverse types of DMTs. Studies examining the impact of DMTs on AIDs in MS have produced conflicting results. Medications that have been demonstrated to induce such adverse effects include interferons, β-cell depletion therapies, and biologic agents [44,45]. Emerging evidence indicates a link between alemtuzumab use and the development of thyroid disorders [46,47,48]. Contrary, in a study by Sovetkina et al., a decrease in EDSS score was observed among patients who developed autoimmune thyroiditis (AITD) after receiving alemtuzumab [49]. Interferons may exacerbate existing autoimmune processes or induce their development de novo. Cases of psoriasis and RA have been reported to emerge or worsen following interferon therapy, leading to recommendations against using interferons in patients with pre-existing AIDs [44,49,50,51,52]. Conversely, certain immunomodulatory therapies for MS may positively influence the course of coexisting AIDs. Appropriate drug selection could facilitate the simultaneous management of both conditions. For example, natalizumab has been associated with exacerbations of psoriasis, onset of RA, and thyroiditis [53,54,55]. Nevertheless, in PwMSs and Crohn’s disease, natalizumab has been effective in controlling disease progression and maintaining remission [56,57]. Ozanimod represents a pharmaceutical agent that has been clinically proven to be effective in the treatment of both MS and ulcerative colitis (UC) [58,59]. Other medications, such as teriflunomide and leflunomide, demonstrate positive effects on specific AIDs, such as RA or psoriasis. This underscores the importance of carefully choosing therapies for patients with MS and coexisting AIDs [60]. The outcome observed in our study may be attributed to the relatively considerable number of PwMSs with thyroid disorders, for which most DMTs do not have a proven exacerbating effect. It is, therefore, possible that these opposing actions balanced each other in our study, leading to results that suggest no significant influence, or that any potential impact of DMTs is not substantial enough to be detected in this cohort. The relationship between DMTs and AIDs in MS undoubtedly requires further investigation due to the inconsistencies in the literature.

### 4.3. Co-Occurrence of MS and AIDs

In our study group, the prevalence of most AIDs was found to be like that observed in the general population [61,62,63,64,65,66,67,68]. Rheumatic diseases such as psoriatic arthritis (prevalence 2–3% in European population), RA (0.51–0.56% in international retrospective studies), and ankylosing spondylitis (0.13–0.18% in ankylosing spondylitis patients) occur at rates comparable to those observed in the general population [61,62,63]. Similarly, skin conditions, including vitiligo (global prevalence 0.5–2%), atopic dermatitis (frequency in Europe 17.1%), and psoriasis (in the Polish population 2.99%), showed similar incidence rates [64,65,66]. No significant deviations from general epidemiological trends were observed for inflammatory bowel disease (approximately 0.16% prevalence in the Polish population) or type 1 diabetes (9.5% frequency in the general population) [67,68]. The prevalence of Sjögren’s syndrome was consistent with the rates observed in the general population, ranging from 0.2% to 0.1% [69].

The only category of diseases that showed a higher prevalence in our cohort compared to the general population were thyroid disorders and unspecified thyroid dysfunctions related to hyperthyroidism (0.2–1.3% in the general population) and/or hypothyroidism (0.2–5.3% in the European population) [70,71]. The exception is Hashimoto’s disease (5% in general population), which suggests that major percentage of hypothyroidism in PwMSs have other underlying factors [70,71,72].

### 4.4. Thyroid Disorders Are the Most Common AIDs in PwMSs

In our study cohort of PwMSs, the most associated AIDs were thyroid disorders, affecting 13.28% of patients, which was higher than the prevalence in the general population [71,72,73,74,75,76]. The most frequently diagnosed condition was hypothyroidism, observed in 7.24% of PwMSs. The next most prevalent thyroid AIDs was Hashimoto’s disease, found in 4.18% of patients. These findings are consistent with those reported by Wawrzyniak et al., where 3.7% of all PwMSs included in the study were also diagnosed with Hashimoto’s disease [37]. Hyperthyroidism was reported by 1.72% of individuals with MS. The literature also indicates that PwMSs are more likely to have thyroid disorders compared to the general population, and thyroid diseases affect up to 17.1% of PwMSs, suggesting a significant co-occurrence of thyroid dysfunction within this group [37,77,78]. This highlights a noteworthy association between MS and thyroid disorders in clinical populations.

Based on the collected data, women with MS are more than twice as likely as men with MS to suffer from concurrent thyroid disorders. This finding is corroborated by studies where authors also reported a higher incidence of thyroid diseases among female PwMSs, whereas the results for male PwMSs were statistically insignificant [79,80].

Analysis of thyroid dysfunction in PwMSs conducted by Poursadeghfard et al. revealed that abnormal TSH levels were present in newly diagnosed PwMSs, even before the initiation of treatment [79]. In a study conducted by our team, which included patients after MS diagnosis and the initiation of relapse-modifying and DMTs, results indicated that most patients with diagnosed AIDs had abnormal TSH levels. The largest group comprised patients with elevated TSH levels, suggesting that individuals with MS may be at increased risk for thyroid dysfunction [79]. The precise cause of this phenomenon remains to be definitively established, necessitating further research to elucidate it. There is some evidence to suggest that the utilization of alemtuzumab and interferon beta may be contributory factors in this regard [80,81].

### 4.5. Role of Vitamin D3 in the Pathogenesis of MS and AIDs

Vitamin D3 deficiency is considered to be one of the risk factors for MS, and the supplementation of vitamin D3 has been extensively studied as a form of adjunctive therapy for MS. However, to date, it has not been conclusively demonstrated that vitamin D3 supplementation significantly affects the clinical course or progression of MS [24]. The findings of the research indicate that 47.59% of the participants exhibited reduced levels of vitamin D3. In addition, a growing body of research has identified vitamin D3 deficiency as a contributing factor in the pathogenesis of autoimmune thyroid diseases, promoting their development and progression. In contrast, in other AIDs such as SLE, thyrotoxicosis, type 1 diabetes, Crohn’s disease, UC, psoriasis vulgaris, and seropositive RA, no significant correlation has been established between vitamin D3 levels and the risk of developing these conditions [25]. Our findings seem to support these conclusions, as no significant differences in vitamin D3 levels were observed between PwMSs with AIDs and PwMSs without coexisting AIDs.

### 4.6. Limitations and Implications

It is important to consider the limitations of this study, including its comparative retrospective nature, which limits the ability to infer causality and to account for potential selection bias. Another limitation of the study is that it was conducted at a single center, involving patients from only one region of Poland. Notably, to the best of our knowledge, this represents the most extensive study on this subject conducted in Poland to date. It is regrettable that the study did not permit an evaluation of the timing of AIDs onset. Patients with pre-existing AIDs were not distinguished from those who developed these conditions only after initiating immunomodulatory treatment. This represents a significant limitation, as it has been demonstrated that many drugs used in MS therapy can either induce the development of AIDs or, conversely, positively impact their clinical manifestation.

## 5. Conclusions

The most notable observation derived from our investigation is that 16.9% of PwMSs presented with a concomitant autoimmune disorder. The most common were autoimmune thyroid disorders including hypothyroidism, Hashimoto’s disease, and hyperthyroidism. A comparison of patients with MS and those with MS and AIDs reveals several notable differences. Firstly, the average age of patients with MS is higher than that of patients with MS and AIDs. Secondly, the average duration of disease is longer in patients with MS than in patients with MS and AIDs. Thirdly, PwMSs typically present at a more advanced stage of the disease, as indicated by EDSS. Both MS and MS with AIDs are more prevalent in women than in men, which aligns with the trend indicating greater susceptibility of women to autoimmune conditions, particularly thyroid disorders. The prevalence of AIDs remained consistent across PwMSs treated with different forms of DMTs. Also, vitamin D3 levels are not associated with an increased incidence of AIDs in PwMSs. Given the complexity of interactions between MS and AIDs, further studies are essential to elucidate the risk factors, interrelationships, and their effects on disease progression and treatment response. In the context of caring for people diagnosed with MS, particular attention should be paid to thyroid disease, which our analysis has identified as being especially prevalent within this group.

## Figures and Tables

**Figure 1 brainsci-15-00588-f001:**
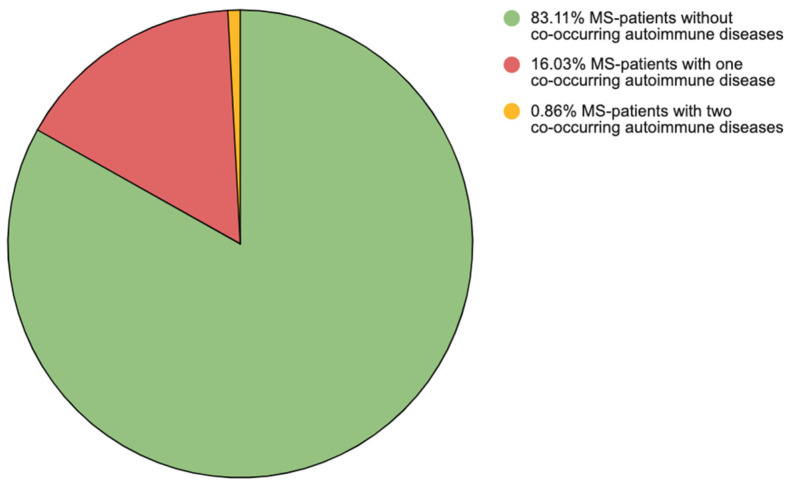
Division of multiple sclerosis patients according to the number of co-existing autoimmune diseases. Note: MS—multiple sclerosis.

**Figure 2 brainsci-15-00588-f002:**
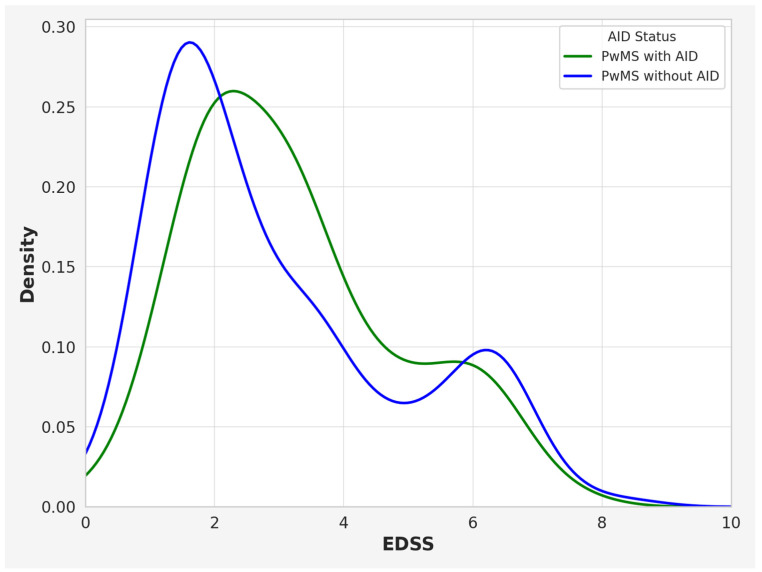
Kernel density plot illustrating the distribution of EDSS scores in PwMSs with and without AIDs. Note: EDSS—Expanded Disability Status Scale; PwMSs—patients with multiple sclerosis; AIDs—autoimmune disease.

**Figure 3 brainsci-15-00588-f003:**
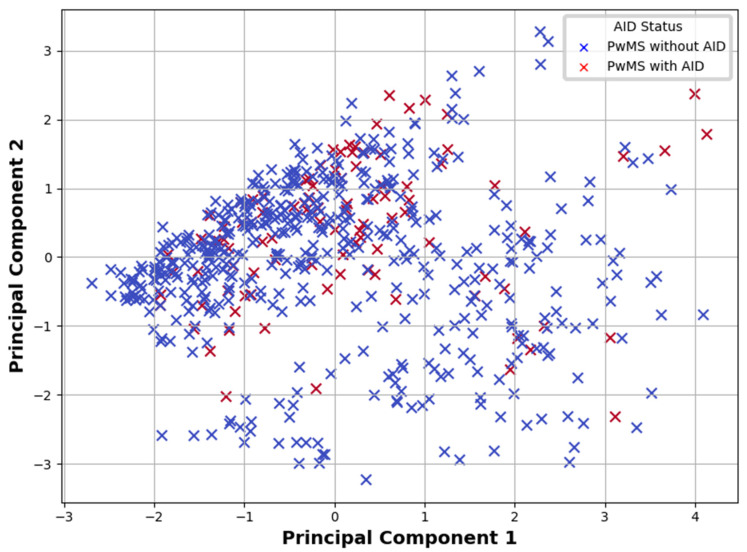
Principal Component Analysis plot showing the distribution of PwMSs with and without AIDs based on the first two principal components derived from clinical data. Note: PwMSs—patients with multiple sclerosis; AIDs—autoimmune diseases.

**Table 1 brainsci-15-00588-t001:** General description of the study group.

Characteristic	
Total number of PwMSs	580
Sex, n (%)	
– Female	431 (71.71%)
– Male	170 (28.29%)
Age, mean ± SD (range)	42.88 ± 12.56 (range: 18–73)
Age at MS diagnosis, mean ± SD	33.63 ± 11.22
Disease duration, mean ± SD	9.82 ± 7.58
MS Phenotype, n (%)	
– RRMS	469 (80.86%)
– SPMS	48 (8.28%)
– PPMS	44 (7.58%)
– RES-RRMS	19 (3.28%)
EDSS score, median ± IQR	2.50 (IQR: 1.25–3.75)
Treatment Status, n (%)	
– LETA	301 (51.90%)
– HETA	216 (37.24%)
– No treatment	63 (10.86%)
Most Common DMTs, n (%)	
– Dimethyl fumarate	168 (28.97%)
– Teriflunomide	64 (11.03%)
– Ocrelizumab	60 (10.34%)
– Ofatumumab	27 (4.66%)

MS—multiple sclerosis; RRMS—relapsing-remitting MS; SPMS—secondary progressive MS; PPMS—primary progressive MS; RES-RRMS—rapidly evolving severe RRMS; EDSS—Expanded Disability Status Scale; LETA—low-efficacy treatment approach; HETA—high-efficacy treatment agents; DMTs—disease-modifying therapies; PwMSs—patients with multiple sclerosis.

**Table 2 brainsci-15-00588-t002:** Distribution of vitamin D3 and TSH serum levels in patients with multiple sclerosis, stratified by normal, low, and high ranges.

Measurement	Mean ± SD	Below Normal (n, %)	Normal Range (n, %)	High/Toxic (n, %)
Vitamin D3 serum level	34.52 ± 27.20 ng/mL	237 (47.59%)	250 (50.20%)	11 (2.29%)
TSH level	2.06 ± 4.54 µIU/mL	12 (2.42%)	469 (94.75%)	14 (2.83%)

Note: SD—standard deviation; vitamin D3—25-hydroxyvitamin D; TSH—thyroid stimulating hormone.

**Table 3 brainsci-15-00588-t003:** The co-occurrence of autoimmune diseases and the prevalence of thyroid diseases in multiple sclerosis subtypes.

MS Subtype	Patients with Autoimmune Comorbidities	Patients with Co-Occurring Thyroid Disease
RRMS	18.12%	14.29%
SPMS	16.67%	14.58%
PPMS	11.36%	6.82%
RRMS-RES	0.0%	0.0%

Note: PPMS—Primary Progressive Multiple Sclerosis; RRMS—Relapsing-Remitting Multiple Sclerosis; RRMS-RES—Rapidly Evolving Severe RRMS; SPMS—Secondary Progressive Multiple Sclerosis.

## Data Availability

The datasets presented in this article are not readily available because of the privacy of the patients and the anonymization process. Requests to access the datasets should be directed to the corresponding author.

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
