# Peer review of "The Relationship Between Autoimmune Disorders and Multiple Sclerosis: Clinical Insights and Therapeutic Approaches"

_brainsci, 2025, doi:10.3390/brainsci15060588_

Round 1
Reviewer 1 Report
Comments and Suggestions for Authors
METHODS
Study design mislabelled. The manuscript calls the work a “retrospective cross-sectional study” (l. 231 ff.), yet exposure (presence of AID) and outcome (EDSS, disease duration, etc.) are all measured on or before the same index date; moreover, comparisons are between groups, not across time. A more accurate description is a retrospective cohort (comparative) study.
Group differences are assessed only with univariate tests; confounding by age, sex, and disease duration—known determinants of EDSS—cannot be excluded. I recommend running a multivariable linear (or ordinal) regression for EDSS and logistic regression for the presence of AID, including at least age, sex, MS phenotype, vitamin D₃ level, and treatment class.
This study needs to better identify the term thyroid disease. If referring to an autoimmune thyroid disorder, state it as such. There should also be a breakdown of what thyroid disease was seen in the patients, e.g., Graves Disease, Hashimoto's, etc.
RESULTS
The EDSS is being treated as normally distributed. Mean ± SD is reported, yet EDSS is bounded (0–10) and skewed. I recommend reporting median [IQR] and using Mann-Whitney U or ordinal regression; provide kernel density/histogram.
The missing-data handling is not described. Inclusion was limited to “complete clinical data” (l. 94-97), but the proportion excluded and the reasons are absent.
Author Response
Dear Reviewer,
We would like to thank you for the review of our manuscript, which helped us improve the quality of the paper. In response, we have made the following changes:
Comments 1: METHODS. Study design mislabelled. The manuscript calls the work a “retrospective cross-sectional study” (l. 231 ff.), yet exposure (presence of AID) and outcome (EDSS, disease duration, etc.) are all measured on or before the same index date; moreover, comparisons are between groups, not across time. A more accurate description is a retrospective cohort (comparative) study.
Response 1: Thank you for your comment. We have corrected the study design description to a retrospective cohort (comparative) study.
Comments 1: Group differences are assessed only with univariate tests; confounding by age, sex, and disease duration—known determinants of EDSS—cannot be excluded. I recommend running a multivariable linear (or ordinal) regression for EDSS and logistic regression for the presence of AID, including at least age, sex, MS phenotype, vitamin D₃ level, and treatment class.
Response 2: Thank you for this valuable suggestion. In response, we conducted multivariable logistic regression for AID presence and linear regression for EDSS, including age, sex, MS subtype, disease duration, and treatment status as predictors. Results confirmed that age and progressive MS subtypes were independently associated with both AID occurrence and higher EDSS. These findings are now included in the revised manuscript (lines 293-320).
Comments 3: This study needs to better identify the term thyroid disease. If referring to an autoimmune thyroid disorder, state it as such. There should also be a breakdown of what thyroid disease was seen in the patients, e.g., Graves Disease, Hashimoto's, etc.
Response 3: We thank the Reviewer for highlighting this important point. We have clarified in the manuscript that the term "thyroid disease" specifically refers to autoimmune thyroid disorders. (Lines 213-216)
“The most prevalent autoimmune disorders observed were autoimmune thyroid diseases, present in 77 PwMS (13.28%), including hypothyroidism (7.24%; n = 42) and Hashimoto’s disease (4.18%; n = 24), hyperthyroidism (1.72%; n = 10) and other/unspecified thyroid disorders (0,17%; n=1).”
Comments 4: RESULTS. The EDSS is being treated as normally distributed. Mean ± SD is reported, yet EDSS is bounded (0–10) and skewed. I recommend reporting median [IQR] and using Mann-Whitney U or ordinal regression; provide kernel density/histogram.
Response 4: Thank you for this remark. While the Mann–Whitney U test was already used for group comparisons of EDSS, we have revised the reporting format to better reflect the ordinal and skewed nature of this variable. EDSS is now presented as median [IQR], and we have added a kernel density plot to illustrate its distribution by AID status, as recommended (Figure 2).
Comments 5: The missing-data handling is not described. Inclusion was limited to “complete clinical data” (l. 94-97), but the proportion excluded and the reasons are absent.
Response 5: In the revised manuscript, we have now clearly described our approach to handling missing data, specifying the proportion of patients excluded (due to incomplete clinical data) along with explicit reasons for their exclusion. From the initial cohort of 840 patients, 260 were excluded due to missing key information, such as confirmed MS diagnosis, EDSS score, MS type, disease duration, or the presence of comorbidities. (lines 124-127).
Reviewer 2 Report
Comments and Suggestions for Authors
The relationship between autoimmune disorders and MS is an important topic, and the presented data are large and very useful. I have suggestions for improvements in the enclosed document.

Author Response
Dear Reviewer,
We would like to thank you for the review of our manuscript, which helped us improve the quality of the paper. In response, we have made the following changes:
Comments 1: Introduction Consider addressing more clearly these moments for the introduction and discussion:
The pathogenesis of the disease incompletely understood. Multiple sclerosis (MS) is a disabling neurological disorder characterized by symptoms, clinical signs and imaging abnormalities that typically fluctuate over time and vary among individuals (Dobson et al 2019, Milo et al 2020, Kuhlmann et al 2023).
• There are ongoing debates on whether inflammation is the primary trigger of the disease or a secondary response to it, and a debate on the concept of MS as autoimmune disease (Trapp and Nave 2008, Stys et al. 2012, t Hart et al 2021, Stys et al 2024).
• Contrasting reports on traditional T-cell autoimmune paradigm in MS (Greenfield et al. 2018, Dobson et al. 2019, Arneth 2024).
• Although MS has historically been considered a demyelinating disease of the CNS and white matter, in recent years, demyelination of the cortical and deep gray matter has been recognized and may exceed white matter demyelination (Kutzelnigg et al. 2005)
Response 1: Thank you for your comment. We have added a section to the introduction and discussion addressing the suggested topics, which states:
- in the introduction: “Multiple sclerosis is widely regarded as an immune-mediated condition; however, the question of whether inflammation is the primary driver or a secondary epiphenomenon remains controversial. Two mechanistic frameworks have emerged in recent years. The classical 'outside-in' paradigm attributes early perivascular T-cell infiltration and demyelination to aberrant adaptive immunity, whereas the 'inside-out' concept proposes that a primary oligodendrocyte or axonal insult triggers a secondary immune response [15,16,17]. Converging high-field MRI and neuropathological studies also demonstrate that demyelination of the cortical and deep grey matter can equal or even exceed white-matter involvement, thus challenging the historical view of MS as a purely white-matter disease [18,19]. It is imperative to reconcile these perspectives in order to interpret the comorbidity patterns, including autoimmune co-occurrence, observed in multiple sclerosis.”
-in the discussion: “The findings of this study indicate a slightly higher EDSS and longer disease duration in PwMS who also have co-existing AID. These findings directly address the long-standing debate over whether inflammation ("outside-in") drives neurodegeneration in MS or represents a secondary response to a primary oligodendrocyte insult ("inside-out"). The modest clinical difference, in conjunction with an unaltered relapse rate, suggests that systemic autoimmunity may be an epiphenomenon rather than a disease-accelerating factor [15,16,17]. In view of the recent evidence for extensive grey-matter demyelination [18,19], our observations highlight the necessity for integrative models that accommodate both mechanisms along the MS continuum.”
Additionally, we incorporated 5 out of the 12 recommended references.
Comments 2: Material and method
Line 103
What is the precise definition applied here on Autoimmune diseases (AID) for the analyses of individuals with coexisting AID and MS. These data are obviously multivariate, yet only univariate data analyses are applied. I would recommend applying multivariate analyses. There are many alternatives for multivariate analyses of these data. My suggestion is first to apply explorative analyses by Principal Component Analyses (PCA) to see the overall structure in the data. And then to apply the supervised method Partial Least Squares (PLS) regression. The input data could be clinical data and the classification as RRMS, SPMS, PPMS and RES- 186 RRMS. The response data could be two classes: class 1 (MS without AID) and class 2 (MS with AID). Note that with PLS, the input data may be correlated. There are different approaches available for feature extraction in PLS, for example Jackknife.
Response 2: We thank the Reviewer for this valuable suggestion. We have clarified the precise definition of autoimmune diseases (AID) used in our study as conditions diagnosed by clinical criteria and confirmed by serological methods (lines 40-43).
Following the Reviewer’s recommendation, we performed exploratory multivariate analysis using Principal Component Analysis (PCA) to evaluate the overall data structure and subsequently applied Partial Least Squares (PLS) regression to differentiate between MS patients with and without coexisting AID. These additional analyses have been included and discussed in the revised manuscript.
Comments 3: Results
Line 170-189
3.3 Comparison between PwMS with and without co-occurring AID
If these results were presented in a table, it would be easier for the reader to see the overview.
Response 3: Figure 2 has been replaced with table 3 (line 248).
Comments 4: Discussion
Line 237-239
The text now says: “In our study, a considerable portion of PwMS were affected by AID (98 out of 580 237 individuals; 16.90%)”. I am not sure if this can be called “considerable proportion”. It would stand better to say: “In our study, a significant portion of PwMS….”. Although it may seem to give the same information, “significant” refers more objectively to the statistical results, whereas “considerable” is vaguer, and include interpretation of its importance in biological means.
Response 4: We made a suggested change in the Discussion on the mentioned line (line 332).
Additionally, the manuscript has undergone thorough English editing to enhance clarity, grammar, and overall readability.
Round 2
Reviewer 2 Report
Comments and Suggestions for Authors
Important improvements have been made, and the study is now well analysed and well written.
There are some minor errors, for example, in the following lines, there is an extra space or a missing space.:
line 39
line 108
line 327
line 387
line 388